**www.cambridge.org/ext**

## Overview Review

mass extinction; background extinction; sepkoski compendium; paleobiology database; temporal resolution; incompleteness of the fossil record; sequence stratigraphy

**Author for correspondence:**
Charles R. Marshall,
Email: crmarshall@berkeley.edu

# Forty years later: The status of the "Big Five" mass extinctions

Charles R. Marshall ⬤

Department of Integrative Biology and University of California Museum of Paleontology, University of California, Berkeley, CA, USA

## Abstract

Over 40 years ago, Raup and Sepkoski identified five episodes of elevated extinction in the marine fossil record that were thought to be statistically distinct, thus warranting the term the "Big Five" mass extinctions. Since then, the term has become part of standard vocabulary, especially with the naming of the current biodiversity crisis as the "sixth mass extinction." However, there is no general agreement on which time intervals should be viewed as mass extinctions, in part because the Big Five turn out not to be statistically distinct from background rates of extinction, and in part, because other intervals of time have even higher extinction rates, in the Cambrian and early Ordovician. Nonetheless, the Big Five represent the five largest events since the early Ordovician, including in analyses that attempt to compensate for the incompleteness of the fossil and rock records. In the last 40 years, we have learned a great deal about the causes of many of the major and minor extinction events and are beginning to unravel the mechanisms that translated the initial environmental disturbances into extinction. However, for many of the events, further understanding will require going back to the outcrop, where the patchy distribution of environments and pervasive temporal gaps in the rock record challenge our ability to establish true extinction patterns. As for the current biodiversity crisis, there is no doubt that the rate of extinction is among the highest ever experienced by the biosphere, perhaps the second highest after the end-Cretaceous bolide impact. However (and fortunately), the absolute number of extinctions is still relatively small – there is still time to prevent this becoming a genuine mass extinction. Given the arbitrariness of calling out the Big Five, perhaps the current crisis should be called the "incipient Anthropocene mass extinction" rather than the "sixth mass extinction."

## Impact statement

We are currently in the beginning stages of the so-called "sixth mass extinction," with rates of species loss that are frighteningly high, even when compared with the highest rates measured in the fossil record over the last half billion years. The term the sixth mass extinction refers to five large extinction events seen in the marine animal fossil record, called the "Big Five." The Big Five were named because they were thought to represent a different type of extinction in contrast to the pervasive background extinction rates seen in the fossil record. Now, 40 years later, what is the status of the Big Five given the better-known fossil record and with new methods for compensating for its incompleteness? While the Big Five remain among the largest of all extinctions in the marine realm and the largest since the early Ordovician, they are not statistically distinct; there is a continuum of extinction intensities from the largest to the smallest. Thus, the decision to call out five of the biggest extinctions rather than some other number is, in retrospect, somewhat arbitrary. Some events were relatively sudden, while others likely extended over hundreds of thousands of years or longer. In terms of rate, the current loss of biodiversity is perhaps the second fastest experienced by the biosphere in the last half billion years, after the end-Cretaceous mass extinction. However, the total number of species that have already gone extinct is still very small compared with the largest extinction events seen in the fossil record; there is still time to act to prevent a true mass extinction, however, they are defined and counted. Given the arbitrariness of calling out the Big Five, perhaps the current crisis should be called the "incipient Anthropocene mass extinction" rather than the "sixth mass extinction."

## Introduction

The term the "Big Five mass extinctions" (the "Big Five") has become part of paleontology's vernacular, attracting attention through natural history museum websites and popular science venues such as Cosmos, Discover Magazine, The Conversation, and the BBC's Science Focus. The notion of the Big Five has also been immured through the characterization of the current biodiversity crisis as the sixth mass extinction, both scientifically (e.g., Wake and Vredenburg,

2008; Barnosky et al., 2011; Ceballos et al., 2017, 2020) and more generally (e.g., via popular books by Leakey and Lewin [1996] and Kolbert [2014]). While there is no disagreement over the identity of the Big Five, the elevated extinction rates associated with the end-Ordovician, Late Devonian, end-Permian, end-Triassic, and end-Cretaceous, the definition of a mass extinction is fuzzy. This fuzziness renders the term "the sixth mass extinction" in a curious state of limbo despite the consensus on the severity of the current biodiversity crisis. In this review, I outline the discovery, characterization, current status, and future challenges in understanding mass extinctions in the marine fossil record, with a focus on the Big Five recognized by Raup and Sepkoski (1982) over 40 years ago.

The recognition and characterization of the Big Five consist of three relatively distinct historical phases, each enabled (and limited) by the data (and tools) available at the time. Phase 1 constituted the initial identification of periods of elevated extinction based on a literal reading of the fossil record. Phase 2 began with the statistical analysis of Sepkoski's global compendium of family-level data, which led to the identification of the Big Five, followed by analyses of Sepkoski's genus-level data, analyses largely consisting of taking the fossil record at face value. Most of the advances associated with characterizing the Big Five occurred during this phase. Phase 3 was ushered in with the establishment and (continuing) growth of the Paleobiology Database (PBDB; https://paleo biodb.org), designed in part to enable quantification of the incompleteness of the fossil record. Analyses of the PBDB have largely corroborated the conclusions derived from Sepkoski's data, but this need not have been the case. The congruence between the temporal pattern of extinction intensities inferred from the different databases and with different ways of compensating for the incompleteness of the fossil record suggests that we have a robust understanding of extinction intensity through the Phanerozoic in the marine realm, albeit at a coarse temporal resolution and only for taxa relatively well preserved in the fossil record.

Before outlining this history, I discuss four factors that bear on the analysis of mass extinctions: taxonomy; temporal resolution; the databases used; and the incompleteness of the fossil record, and then offer a definition and terminology to help make sense of the different ways that the term mass extinction has been used.

## Factors that bear on the analysis of mass extinctions

### Taxonomy and taxonomic resolution

Biodiversity is typically measured by biologists at the species level. However, most paleontologists are uncomfortable with the species-level taxonomy in the fossil record largely because much of the phenotype is not preserved. Thus, in the marine realm the genus is the standard unit of analysis, and so the discussion below centers on genus-level data. Nonetheless, extinction patterns at higher taxonomic levels also carry important macroevolutionary information (Jablonski, 2007, 2008, 2017a,b). Thus, paleontologists also keep track of higher taxonomic rates of extinction, which, for example, play a role in assessing the ecological impact of extinction events (Droser et al., 2000; McGhee et al., 2013; Muscente et al., 2018).

### Estimating species-level extinction intensity

While extinction intensity is measured at the genus level, there has long been interest in converting observed genus-level extinctions into the equivalent number of species lost. There is no fully secure way to do this, but it is not uncommon to see estimates of the species-level extinction for the largest mass extinction, the end-Permian, as high as 96%, the upper limit of Raup's (1979) estimate based on rarefaction analysis of family and ordinal-level data. At the genus level, extinction intensity of the end-Permian based on Sepkoski's data broken down to the substage level is 56% (Bambach, 2006), or at the coarser stratigraphic stage level is 65% (Raup) or between 56 and 69% (Stanley, 2016). Stanley (2016), in a more sophisticated analysis than Raup's (1979), estimates the corresponding end-Permian species-level extinction to be 81% based on rarefaction from the genus-level data. This is still ferociously high but is not 96%.

### Relationship between species-level extinction and the definition of a mass extinction

Mass extinctions represent time intervals where the extinction rate stands out compared with the extinction rate in the adjacent stages, without any required preset threshold. Thus, while some describe mass extinctions as being intervals with species-level extinction rates of greater than 75% (e.g., see Barnosky et al., 2011) this is simply a retrospective characterization of intervals that have already been recognized based on other criteria. In fact, if Stanley's (2016) recalculation of the species-level extinction for the Big Five is correct, then only the end-Permian would actually meet this criterion.

### Taxonomic noise

As one might imagine, many of the times of first and last occurrences in any database might be in error due to taxonomic misidentification and revision, refinement in stratigraphic ranges, and improvement in the geologic timescale. How severe might those errors be? In the only comprehensive analysis of these types of errors, Adrain and Westrop (2000) reanalyzed Sepkoski's Ordovician and Silurian trilobite genus data. They identified a 70% error rate in the entries for 941 trilobite genera, but, stunningly, the diversity trajectory itself, and the percent change from one interval to the next, was almost indistinguishable from the uncorrected data – it appears that (at least for this group) the substantial error rate constitutes harmless white noise (Adrain and Westrop, 2000). It would appear that meaningful analysis of large-scale extinction patterns does not require a bullet-proof taxonomy.

### Phylogenetic status of the taxa

Most named taxa in the fossil record are "phena" (*sensu* Smith, 2009), consisting of specimens that cluster in morphospace. Thus, many of these taxa are likely to be paraphyletic, named groups that do not include all the descendants of a given common ancestor. This in turn means that their lineages may well have persisted (via anagenesis with sufficient change to be given a different name) after the last appearance of specimens assigned to the given taxon name; the fossil record is likely replete with pseudo-extinction. However, given that most genera consist of more than one species, it also seems likely that the disappearance of paraphyletic genera was nonetheless typically associated with the extinction of at least some species-level lineages. Thus, while paraphyletic taxa likely abound in paleontological databases, it seems unlikely that it distorts the extinction patterns to an appreciable degree, although this has yet to be formally demonstrated (but see Silvestro et al., 2018).

### Temporal resolution: Two approaches

There are two completely different approaches for undertaking large-scale analyses of the fossil record. The first uses the global geological timescale, where times of origination and extinction are

assigned to one of the recognized temporal intervals, typically at the stage level. This has been the standard approach for analyzing Phanerozoic extinction patterns, and its properties are discussed immediately below. I then briefly discuss an approach that offers a 100-fold higher temporal resolution, but has not yet been applied to global Phanerozoic data.

With the standard approach, the temporal resolution of large-scale analyses of the fossil record is limited to the temporal resolution of the global geologic timescale. In the initial analysis of mass extinctions by Raup and Sepkoski (1982), the temporal resolution was 7.4 million years, corresponding to the average duration of the Phanerozoic stages (as defined at the time). To give a sense of how long that is, with a median species longevity of ~2 million years (see Marshall, 2017) and a constant rate of extinction, ~90% of a cohort of species will have become extinct after 7.4 million years. With an average genus duration of ~5 million years, 64% of the cohort will have become extinct in the same period.

Most analyses simply used the timescale directly, using the stratigraphic intervals as defined at the time of analysis. However, there is considerable heterogeneity in the stage durations, which span more than an order of magnitude (from less than 1 million years to almost 20 million years). An alternative approach, developed at the inception of the PBDB, was to combine stages as needed to obtain intervals of approximately equal duration, approximately 10–11 million years. This was the resolution used in the first analyses with the PBDB (Alroy et al., 2001), which continues to be used on occasion (Alroy, 2008; Alroy et al., 2008; Kocsis et al., 2019).

### Refinement of the timescale and its impact

The ongoing refinement of the geological timescale (coordinated by the International Commission on Stratigraphy: https://stratigraphy.org) has enabled increasingly finer temporal resolution, for example, in the use of substages that have an average duration of 3.25 million years (e.g., Bambach, 2006). The increased temporal resolution has been important. For example, in almost all analyses at the stage level, the last two stages of the Permian show elevated extinction rates (Raup and Sepkoski, 1982; Bambach et al., 2004). It has been unclear whether the older of the two peaks is real or just due to the expected smearing back of extinction times from the end-Permian mass extinction given the temporal incompleteness of the fossil record, termed the Signor–Lipps effect (Signor and Lipps, 1982). However, analysis at the substage level (Bambach, 2006) resolves two late Permian extinction peaks, the end-Permian and then two substages earlier a distinct peak in the Capitanian, which has independently been identified as a separate extinction (Rampino and Shen, 2021), although it still unclear what proportion of the stratigraphic endpoints in the Capitanian are due to extinction at that time, and what proportion are due to the Signor–Lipps effect from the end-Permian extinction. In this context, the extinction peak is barely visible in the analysis of the PBDB after correction for the incompleteness of the fossil record (Kocsis et al., 2019), and does not appear in the high-resolution CONOP analysis of data from China (Fan et al., 2020) (see below).

### Higher stratigraphic resolution could substantially change our understanding

The temporal resolution of the standard geological timescale is very coarse compared to microevolutionary timescales – for many marine invertebrates a million years represents in the order of a million generations. What would the Phanerozoic extinction dynamics look like if we had much higher stratigraphic resolution? That is, how are extinctions within the stages or substages distributed?

There is some evidence that, on average, extinctions are concentrated at the end of stages (Foote, 2005; Kocsis et al., 2019), suggesting that the biosphere is generally quiescent, punctuated by frequent extinction pulses. While this conclusion is derived from indirect evidence from synoptic datasets, high-precision data can be obtained from individual localities, where this supposition gains support. For example, for several Cambrian stage boundaries, the extinctions occurred in rapid pulses associated with the flooding of anoxic waters onto shallow trilobite-rich shelf waters, the well-known biomere extinction events (Palmer, 1984, 1998; Saltzman et al., 2015). Thus, if the global stratigraphic resolution were 10 times higher, then most of the new Cambrian "microstages" would have low extinction rates, punctuated by microstages with vastly higher rates of extinction.

There are many other stratigraphic intervals, including the end-Permian, end-Triassic, and end-Cretaceous where data from local geologic sections indicate that the extinctions are concentrated at the end of the stages. These data, in combination with mechanistic models of the rates of environmental perturbation that triggered the extinctions, have been used to estimate the true extinction rates (e.g., see Barnosky et al., 2011). Thus, these rates are much higher than the rates estimated using stage-level data, where the calculated rates assume the extinctions occurred over the entire duration of the stage. To pick just one example of the difference in estimating extinction rates from stage-level data and data from local geological sections, the last stage of the Triassic may be as long as 7 million years (Caruthers et al., 2022), yet the anoxic crisis driven by the emplacement of the Central Atlantic Magmatic Province (CAMP) appears to have lasted less than 1% the duration of the stage, only 50,000 years (He et al., 2020). Thus, the true extinction rate was likely ~100 times higher than the extinction rate derived from the global stage-level data. Hence, we cannot meaningfully compare rates of extinction of taxa in the PBDB with true extinction rates, including current rates, as is occasionally attempted (e.g., Neubauer et al., 2021). Below, I return to the issue of the resolution of the stratigraphic record, both in terms of analyzing mass extinction in local geologic sections and in the context of the current biodiversity crisis.

### Can the temporal resolution be too high?

To detect an extinction event the timescale of analysis has to be on the same timescale or longer than the duration of the event. Thus, use of stage-level temporal resolution is suitable for detecting changes in diversity that occurred in less than a stage. Conversely, if the event is long with respect to the timescale of measurement, the event may not be detected, appearing only as a long-term decline smeared out over many time intervals. Thus, for example, in Hoyal Cuthill et al.'s (2020) analysis of the coupling of origination and extinction dynamics with a high resolution of one million years, the Late Devonian extinction(s) barely registers. But it appears (see below) that the Late Devonian taken more broadly represents one of the biggest extinctions, with dramatic and sustained changes in marine ecosystems (McGhee et al., 2013; Muscente et al., 2018), so the absence of the Late Devonian in Hoyal Cuthill et al.'s (2020) tabulations does not mean it is not a mass extinction.

### An approach with 100-fold higher temporal precision

The standard approach to measuring diversity change consists of assigning times of first and last occurrences to the time intervals in the geologic timescale. However, there is another approach that

does not use the standard geologic timescale. It is called constrained optimization (CONOP), and it orders the first and last occurrences of all taxa derived from multiple measured sections calibrated with geochronologic data (when available) derived from those same outcrops, typically radiometric dates (Sadler and Cooper, 2008). A measured section constitutes rock outcrop level data where the order of appearance and disappearance of taxa are recorded. The largest scale analysis to date is a Cambrian to Triassic Chinese data set from the Geobiodiversity Database (GBDB: http://www.geobiodiversity.com) with an interpolated temporal resolution of 0.026 ± 0.0149 million years, a 100-fold higher than simply assigning last occurrences to their appropriate geologic stage (Fan et al., 2020). The analysis was based on 3,112 measured sections for 11,268 species, enabled by parallelizing the original CONOP simulated annealing algorithms (Sadler, 2004; Sadler et al., 2011). Without this advance, the analysis would have taken dozens of years (Fan et al., 2020). I suspect that the broader application of this approach will revolutionize the temporal resolution of large-scale paleontological data.

## Data sources

All analyses of Phanerozoic extinction have been based on compilations derived from the literature, not yet able to make use of paleontology's substantial dark data, the unpublished data in museum collections (Marshall et al., 2018), or outcrop-level data (Fan et al., 2020). The analyses have been largely based on Sepkoski's compendia and on the PBDB.

### Sepkoski's compendia

The first analysis used Sepkoski's global family-level compendium (Sepkoski, 1992), which was quickly superseded by analyses with Sepkoski's genus-level compendium (Sepkoski, 2002). The genus-level compendium was a 20-year effort in the library as Sepkoski worked systematically through the literature adding taxa and updating taxon names and times of first and last occurrences to existing compilations. Thus, the compendium represented an essentially complete synopsis of the world's paleontological literature at the time. Almost all the advances in understanding the general properties of mass extinctions were based on Sepkoski's genus-level compendium. Nonetheless, the compendia themselves (Sepkoski, 1992, 2002) now represent "frozen" legacy data sets, based on the taxonomies and geological timescale as they were understood at the time, although Heim et al. (2015) updated the taxonomy of the genus database using the PBDB to update synonymies and changes in rank.

### The PBDB

Despite the monumental effort, Sepkoski's compendia are relatively data poor; they simply consist of lists of taxon names and their times of first and last occurrence. There are no data on any of the stratigraphically intermediate occurrences, thus no data on how rich or poor the fossil record is for each taxon. Nor are there geographic or litho-stratigraphic data, or information on the tectonic setting of the fossiliferous rock units, etc. We (John Alroy and I) initiated the PBDB to make these data web-accessible – the PBDB is collections based, where for each published fossil collection (locality), the taxa present, its location, available stratigraphic, taphonomic, tectonic data, etc., can be entered. Thus, the PBDB affords the opportunity to measure collection intensity (e.g., the number of occurrences for each taxon temporally and spatially), making it possible to standardize sampling when assessing marine

diversity dynamics (Miller and Foote, 1996; Alroy et al., 2001; Alroy, 2008; Close et al., 2018; Kocsis et al., 2019).

Now, some 20 years later, the PBDB has a substantial amount of data and has become a standard starting point for large-scale analyses of the fossil record. At the time of writing, there were over 1.5 million occurrences from 225,000 fossil collections derived from 82,000 publications.

### How comprehensive is the PBDB?

Sepkoski's compendia represent relatively faithful representations of the published literature at the time of their compilation. Is the PBDB similarly representative? Much of the data in the PBDB has been entered piecemeal by over 410 paleontologists, each with different priorities and time available. Thus, unlike Sepkoski's compendia which we know were relatively taxonomically complete, we do not know if this is yet true of the PBDB. The fact that the rate of description of new taxa continues at a considerable pace, and that the most recent analysis of the PBDB in the context of the Big Five (Kocsis et al., 2019) used slightly less genera than are documented in Sepkoski's genus compendium suggest the PBDB is not yet a complete representation of the literature. It would be fruitful to develop a way of determining how completely the PBDB reflects the literature. Thus, despite the age of Sepkoski's genus compendium, it is still a useful benchmark even as the PBDB continues to grow.

### The temporal incompleteness of the fossil record

The fossil record is temporally incomplete, with last occurrences being older than true times of extinction (e.g., see Marshall, 2010). Moreover, sampling intensity is uneven, environmentally, temporally, and spatially (Raup, 1972; Peters, 2008; Close et al., 2020). Some stratigraphic intervals and environments are represented by more fossiliferous rock (Raup, 1972; Peters, 2008) and have been searched more comprehensively. Given these issues, to what extent does taking the fossil record at face value distort our view of Phanerozoic extinction rates, a problem that has long been recognized (Newell, 1952; Raup, 1972; Hallam and Wignall, 1999; Smith and McGowan, 2007; McGowan and Smith, 2008; Peters, 2008; Close et al., 2020, among many others)? The temporal incompleteness means that extinction pulses will be smeared back in time (the Signor–Lipps effect), thus diluting the observed intensity of extinction pulses (e.g., see Foote, 2003; Lu et al., 2006, among others), although this aspect of the incompleteness of the fossil record is unlikely to create spurious extinction peaks. But it will affect measured extinction intensities (Foote, 2003, 2007; Lu et al., 2006), so may impact which intervals are identified as times of high extinction.

Initially, there was no way of dealing with the Signor–Lipps effect or the temporal and spatial heterogeneity in preservation potential and fossil recovery, and so most of the literature on the Big Five predates the ability to assess the impact of these factors on the veracity of the Big Five. However, Foote (2003) provided a method to correct the observed rates of extinction (and origination) derived from Sepkoski's genus-level compendium for the incompleteness of the fossil and rock records (see below), while Alroy (2008) and Kocsis et al. (2019) have undertaken analyses of the PBDB data using a variety of methods for trying to deal with the complex issues of temporal incompleteness, heterogeneous preservation, uncertainties associated with estimating model parameter values, and uneven documentation of the fossil record. Crucially, while there are differences in the perceived patterns of extinction before and

after correcting for the incompleteness of the fossil record, the broadest generalities have remained intact (see below).

## A definition and classification of mass extinctions

### Definition of a mass extinction

Definitions can be insidious, where one can almost always find deficiencies in any definition. Nonetheless, Sepkoski (1986, p. 278) provides a qualitative definition that I suspect would meet with broad agreement (e.g., see Bambach, 2006): "*A mass extinction is any substantial increase in the amount of extinction (i.e., lineage termination) suffered by more than one geographically wide-spread higher taxon during a relatively short interval of geologic time, resulting in an at least temporary decline in their standing diversity.*" This definition incorporates the ideas that a mass extinction has a higher extinction intensity compared to the intensities in the adjacent intervals, that more than one major group must be affected (so the end-Holocene mammalian megafaunal extinction is not a mass extinction), and that they involve more than just long-term turnover of taxa.

### A classification of mass extinctions

There are two primary considerations relevant to the identification of mass extinction time intervals. First, whether the time intervals have extinction rates that are significantly distinct from the extinction rates in the other time intervals in the analysis. Second, whether all or just a subset of time intervals were used in the analysis. Combined, these two considerations lead to four types of mass extinction (Table 1).

Note that this classification is based on the analysis of differences in magnitude, not on differences in cause or effect (Wang, 2003) between background and mass extinctions. Also note that in the absence of any significance threshold the number of time intervals identified as Type 3 and Type 4 (Table 1) mass extinctions is arbitrary. I now turn to the history of our understanding of the Big Five mass extinctions.

## Phase 1: On the discovery of times of elevated extinction

The recognition of elevated extinction rates dates back to Phillips (1860) in his analysis of Morris' (1843) compilation of the British fossil record, where two dramatic turnovers in the dominant higher taxa had already led to the demarcation of the Paleozoic, Mesozoic and Cenozoic Eras. Newell (1952) reinitiated the large-scale study of diversity change through time, recognizing similarities in the diversity trajectories of the major fossiliferous groups, thus arguing that they were responding to a set of common environmental

**Table 1.** Classification of mass extinction types

|  | Time intervals used in comparison | |
| --- | --- | --- |
| Extinction rates statistically distinct? | All | Some |
| Yes | Type 1 | Type 2 |
| No | Type 3 | Type 4 |

*Note:* Types 1 and 2 represent time intervals with statistically distinct extinction rates compared with background rates when all other time intervals are analyzed (Type 1) or just a subset of intervals are analyzed (Type 2). Types 3 and 4 represent the intervals with the largest extinction rates among a broader continuum of rates when compared to all other intervals (Type 3) or just a subset of those other intervals (Type 4).

causes. Newell (1952) also noted the correlation between the rises and falls of genus richness and the degree of flooding of the continents, and thus the amount of rock available for analysis. Understanding the relationship between the nature of the rock record and the fossil record that it carries still represents a major challenge as we grapple with the causes of extinction in the fossil record (see below). In the context of the Big Five mass extinctions, while the term stemmed from Raup and Sepkoski's (1982) analysis, each of these times of unusually high extinction had already been recognized by the 1960s (Newell, 1962, 1963, 1967).

## Phase 2a: Uncorrected analyses of Sepkoski's compendia

### The initial recognition of the Big Five

The Big Five were first proposed as a distinct group by Raup and Sepkoski (1982) based on their analysis of the distribution of extinction magnitudes of marine families for 76 geologic stages of the Phanerozoic. Their data were derived from Sepkoski's family-level compendium (available to anyone who asked, but finally published in 1992 [Sepkoski, 1992]), which drew on Moore et al.'s (Moore, 1953–1979) multivolume *Treatise on Invertebrate Paleontology*, Harland's (1967) *The Fossil Record*, Romer's (1966) *Vertebrate Paleontology*, and 380 additional papers and monographs.

At the time of their analysis, we knew virtually nothing about the causes of extinction, and so as a first step Raup and Sepkoski (1982) were interested in whether the variation in the extinction intensities could be accounted for with just one distribution and thus one class of explanation, or whether there were different classes of events, background extinction with its set of causes, and mass extinctions with their own set of causes. We now know so much more than we did about many of the extinction events that it has rendered this simple dichotomy obsolete, but that is the context that led to the recognition of the Big Five; as the great vertebrate paleontologist Alfred Romer noted in 1962 at the International Colloquium on the Evolution of Mammals in Brussels, "*in [vertebrate] paleontology, increasing knowledge leads to triumphant loss of clarity.*"

Using the total number of family-level extinctions per million years, Raup and Sepkoski (1982) found four stages that fell significantly above the 99% confidence interval on the line of best fit of extinction intensity versus geologic time. These four data points corresponded to the last stage of the Ordovician, the last two stages of the Permian, and the last stage of the Cretaceous. A fifth stage and fourth-time interval, the last stage of the Triassic fell above the 95% confidence interval. Finally, the last three stages of the Devonian also fell above the 95% confidence interval, which when taken together made up the complement of the Big Five. The Big 5 as initially proposed constituted Type 1 mass extinctions (Table 1).

As might be expected, the Big Five are seen as the largest drops in diversity in the Phanerozoic family-level diversity trajectory (Figure 1). They are even more evident in Sepkoski's genus-level diversity curve, where the complexity associated with the late Devonian can also be seen (Figure 1).

### An immediate fly in the ointment: The Big Five or just the biggest five?

In their statistical analysis, Raup and Sepkoski (1982) assumed that the extinction rates were normally distributed. However, Quinn (1983) pointed out that the survivorship of a family over its history is best thought of as the product of the probabilities of its survivorship

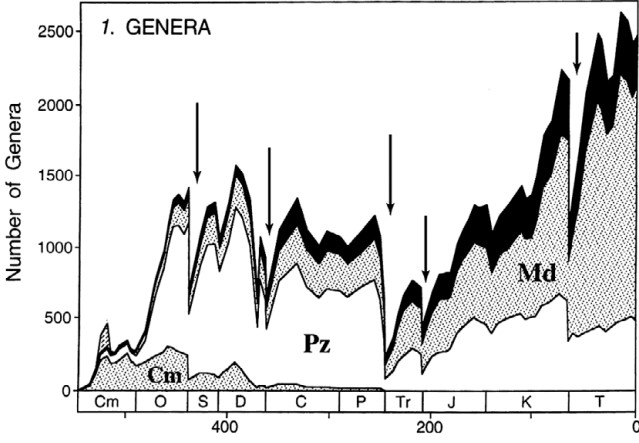

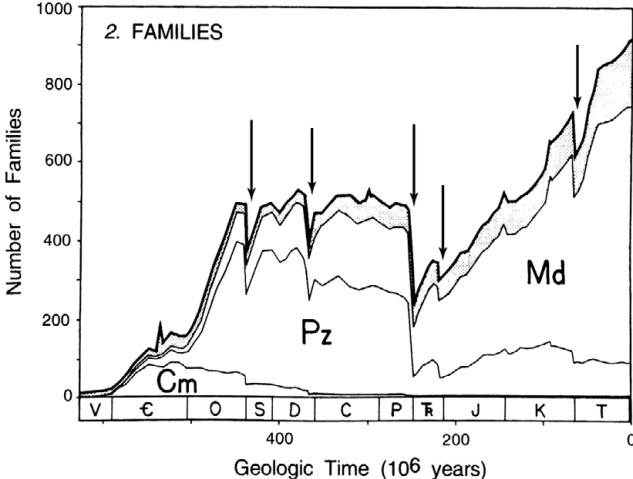

**Figure 1.** Stage-by-stage diversity trajectories of marine genera (top panel) and families (bottom panel) derived from Sepkoski's compendia. The Big Five Mass Extinctions, inferred from an analysis of the extinction rates (data not shown, but see Figures 2 and 3), are indicated by the arrows. They correspond to the times of largest diversity loss. Note that in the genus-level data the Devonian consists of two peaks, indicating that the Late Devonian is more complex than the other big extinctions. Figure modified from Sepkoski (1997).

over successive shorter intervals. This multiplicative property implies the rates should be lognormally distributed, and he showed that, indeed, a lognormal distribution fits the family-level extinction data much better than a normal distribution. Assuming a lognormal distribution he then showed that none of the Big Five stand out as being distinct (Quinn, 1983). Thus, the Big Five were almost immediately demoted from Type 1 to Type 3 mass extinctions and might have been more appropriately called the Biggest Five, with the corollary that there is no special reason to single out the biggest five rather than some other number of time intervals.

### The analysis of the genus compendium

By the time the genus compendium became available, the Big Five had become an established part of paleontological terminology. In 2003, Wang (2003) revisited the issue of whether there was a statistical discontinuity in magnitude between the Big Five and the remaining stages of the Phanerozoic. Using a kernel density estimation technique, he found no evidence of a discontinuity between background and mass extinctions for the genus-level data. Interestingly, a decade earlier Raup (1991) in his kill curve analysis of

Sepkoski's data, also showed that the distribution of extinction intensities can be modeled with a single process, consistent with no fundamental dichotomy between mass and background extinctions.

Nonetheless, Raup (1991) also ended with the cryptic statement that "the fossil record does give a strong intuitive impression of two kinds of extinction, mass extinction, and background extinction." Most paleontologists would agree, but not based on the statistical analysis of extinction intensities, but because the selective regime for most of the largest extinction events appear decoupled from the selective regime during background times (Jablonski, 2001), and with the loss of higher taxa being concentrated at times of mass extinction.

And so, given the deep interest in understanding the biggest extinction episodes in the history of life (at least as recorded in the fossil record), the Big Five became part of regular paleontological terminology, not due to any statistical argument, but simply because they represent the largest diversity drops seen in the Phanerozoic marine fossil record (Figure 1).

### *A second fly in the ointment: A switch in the extinction rate metric*

Raup and Sepkoski (1982) used the total number of families that became extinct in each time interval as their measure of extinction, irrespective of how many families might have been extant at the time. Since then, the standard metric for measuring extinction intensity has become the proportion of taxa going extinct in each interval, rather than the total number, to accommodate the fact that standing diversity is not generally constant. Thus, rates of extinction have the units of number of taxa becoming extinct, per taxon, per million years.

Because Raup and Sepkoski (1982) used absolute rates rather than proportional rates, the low family-level diversity in the Cambrian and early Ordovician (Figure 1) meant that their analysis was not troubled by what turns out to be high proportional rates in the Cambrian (and early Ordovician). Once per capita extinction rates were used and analysis moved to the genus compendium, the Cambrian and early Ordovician dominate the list of stages with the highest per capita extinction rates (Figure 2; Bambach et al., 2004; Bambach, 2006). With per-capita extinction rates, the Big Five are not the biggest five in the Phanerozoic.

It turns out that the origination rates in the Cambrian and early Ordovician were also very high and so the extinction rates are not seen as drops in diversity in the stage-level diversity curves (Figure 1). Thus, with a stage-level temporal resolution, the earliest part of the Phanerozoic is characterized by high turnover rates, a different pattern from the post-early Ordovician, where the Big Five punctuate the remaining portion of the Phanerozoic diversity curve (Figure 1). With the temporal partitioning of the analyses, the Big Five were in effect demoted to Type 4 mass extinctions, the largest five times of extinction in a restricted (but still substantial) portion of the Phanerozoic.

Bambach et al. (2004) also reanalyzed the genus level data for this more limited time series, showing that two stages of the Permian, the end-Ordovician, and end-Cretaceous belong to a different distribution from the rest of the post-early Ordovician stages. Thus, in their analysis, these time intervals were promoted from Type 4 to Type 2 mass extinctions (Table 1). However, this discontinuity in the extinction rate among the post-early Ordovician stages is not seen in any of the analyses that accommodate the incompleteness of the fossil record (see below), and so the Big Five remain Type 4 events.

### Mass depletions versus mass extinctions

Japan is experiencing increased longevities (decreased mortality) (Tsugane, 2021) yet is undergoing population loss (Coulmas, 2007). Why? Because there is an even more pronounced decrease in birth rate, with just one birth for almost two deaths in 2021. Similarly, in the paleontological record, as long as there is extinction, a reduction in origination rate can lead to significant losses of diversity (Bambach et al., 2004; Quental and Marshall, 2013), whimsically called the Entwives effect (Quental and Marshall, 2013).

Thus, Bambach et al. (2004) argued that the Late Devonian and Late Triassic were unusual among the Big Five in that their extinction-driven diversity loss was accentuated by an even larger reduction in origination rate, accounting for two-thirds of the diversity loss (Bambach et al., 2004). Given that diversity loss can be due to more than elevated extinction rates, Bambach et al. (2004) used the term "mass depletion" for these losses of diversity. Interestingly, the loss of diversity driven by a failure to originate will be protracted in time, the time it takes for background (or near background) rates of extinction to bring the diversity down.

However, when the incompleteness of the fossil record is taken into account, the drop in origination rate for both the Late Devonian and end-Triassic disappears (Foote, 2003; Kocsis et al., 2019), suggesting that the end-Devonian and end-Triassic episodes were sharper than implied by Bambach et al.'s (2004) analysis and that the term "mass depletion" is not needed for any of the Big Five.

### Phase 2b: Corrected analysis of Sepkoski's compendia

#### Dealing with the incompleteness of the fossil record does not change the basic pattern

Foote (2003) provides the only attempt to correct the observed rates of extinction (and origination) derived from Sepkoski's genus-level compendium for the incompleteness of the fossil and rock records. The underlying principle of Foote's (2003) analysis is straightforward, even if technically complicated. Simply, Foote (2003) found the "true" stage-by-stage origination, extinction, and preservation rates that returned the closest match between the resulting number of first and last occurrences in each time interval and those in Sepkoski's database. As might be expected, given the Signor–Lipps effect, the extinction pattern is spikier than in the uncorrected data (compare Figure 2 with Figure 3, top panel). However, the basic pattern stays the same – among the biggest peaks are three Cambrian peaks, and after the early Ordovician the Big Five remain the biggest events, with notable additional subsidiary peaks. Foote (2007) found no evidence of two distributions of rates, so in his analysis, the Big Five remained Type 4 mass extinctions.

### Phase 3: The PBDB

Sepkoski's untimely passing in 1999 meant that his genus-level compendium was last updated by him sometime in 1998 or 1999 (Foote, pers. comm.). Thus, it is more than two decades out of date.

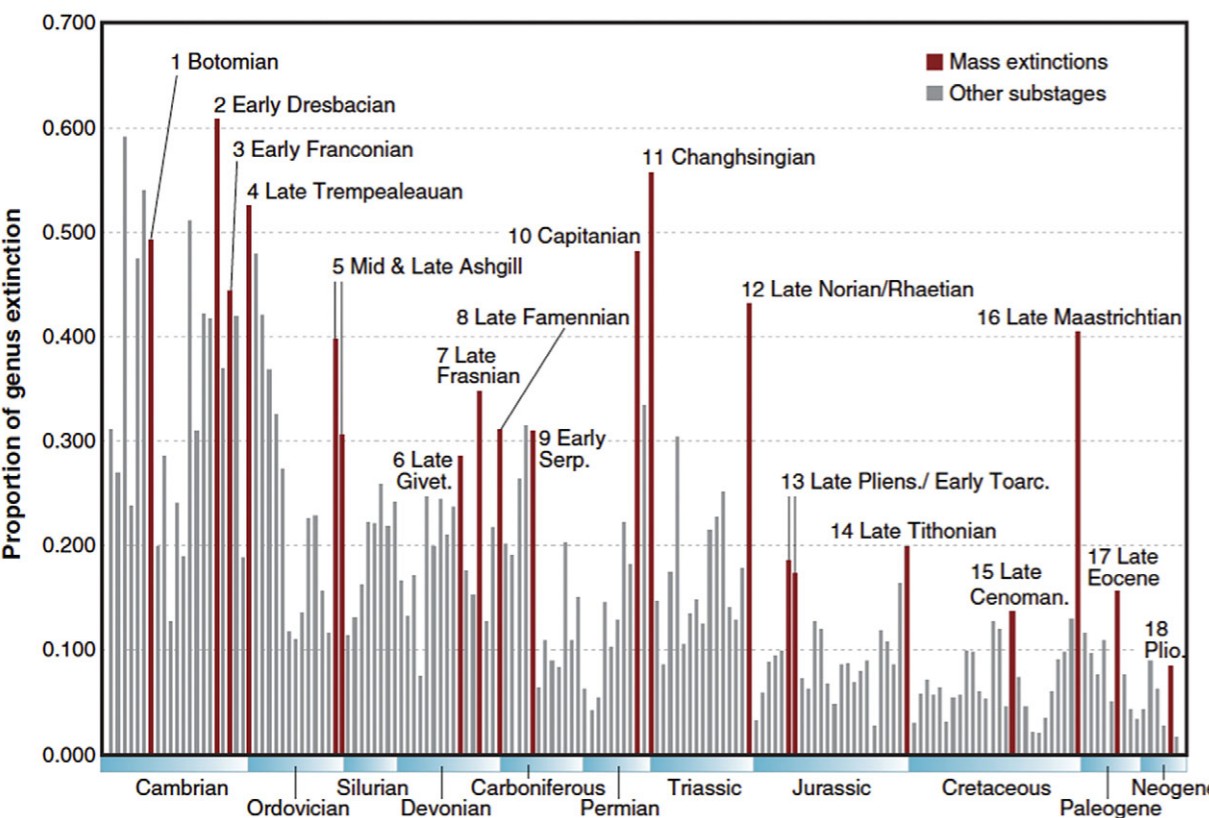

**Figure 2.** Substage-by-substage pattern of extinction intensities of marine genera from Bambach's (2006) analysis of Sepkoski's compendium, showing the 18 local peaks Bambach defined as mass extinctions. The trajectory shows the pervasiveness of extinction inferred from Sepkoski's data prior to attempts to compensate for the incompleteness of the fossil record (Figure 3). The higher temporal resolution used compared with Sepkoski's analyses (Figure 1) suggests that the Capitanian extinction (peak 10) is distinct from the end-Permian extinction (peak 11, the Changhsingian), rather than just being the smearing back of the end-Permian extinction due to the Signor-Lipps effect as was initially thought. Figure reproduced with permission from Bambach (2006) from Annual Reviews.

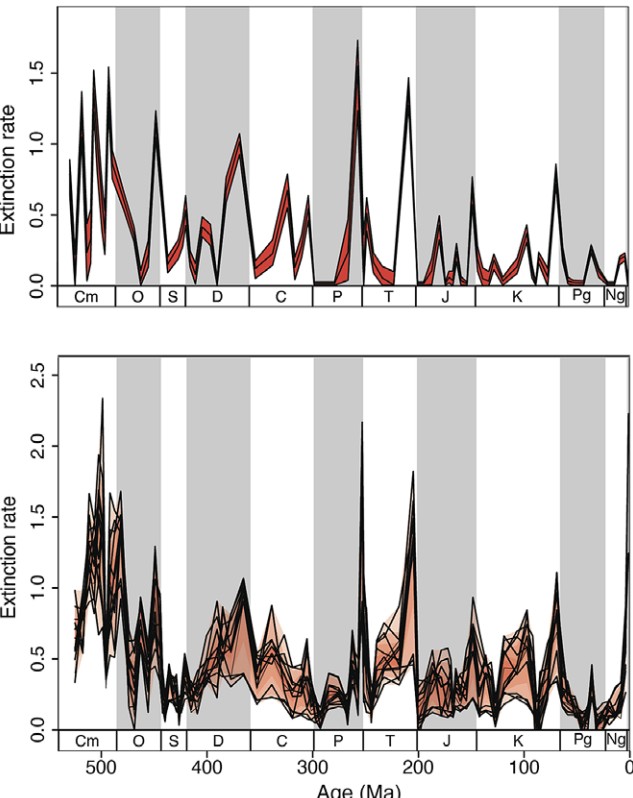

**Figure 3.** Two different methods for compensating for the incompleteness of the marine genus fossil record, each applied to different data sets, yield strikingly similar patterns of stage-bystage marine Phanerozoic per genus extinction rates (genus extinctions/genus/million years). The similarities include the timing and relative magnitudes of the peaks, and the timing of (some of) the stages with zero or close to zero extinction rates. Top Panel: From Foote (2007), based on his analysis of Sepkoski's compendium (Foote, 2003). Bottom Panel: From Kocsis et al. (2019) using the Paleobiology Database. The shading guides the eye to the period boundaries. Ma = millions of years ago.

At about that time the PBDB was being established (we ran the first planning meeting at the National Center for Ecological Analysis and Synthesis [NCEAS] in 1998). In 2008 Alroy (2008) published the first sample standardized analysis of marine origination and extinction rates with the PBDB using the 11-million-year timescale. He confirmed that the post-early Ordovician intervals with the five highest extinction rates correspond to the Big Five, as well as confirming that the extinction intensities were consistent with one distribution of rates: the Big Five remained Type 4 mass extinctions. Note that Alroy's analysis included about 60% of the number of genera used by Sepkoski, ~18,500 genera (but known from ~42,600 fossil collections with ~281,400 occurrences) compared with the ~31,000 genera in Sepkoski's compendium (which also used twice the temporal resolution).

Kocsis et al. (2019) then provided an updated and more comprehensive set of analyses with the PBDB, using ~29,500 genera, just less than the number of genera in Sepkoski's compendium. Reflected in their analysis was the steady refinement of methods for compensating for the incompleteness of the fossil record (Close et al., 2018, see especially Appendix S1 for a detailed discussion of the strengths and assumptions of the Shareholder Quorum Subsampling approach [SQS]) and methods for estimating origination and extinction rates, for example, per capita (PC) rates (Foote, 1999, Appendix 7), corrected three-timer rates (C3t) (Alroy, 2008), the gap filler (GF) approach (Alroy, 2014), and second-for-third (2fer3) rates (Alroy, 2015).

Kocsis et al. (2019) were not focused on extinction, *per se*, but were simply introducing their R package (divDyn) for analyzing diversity dynamics using the sort of data available in the PBDB. Hence, the finer points of their analysis of the Big Five mass extinctions were not included (e.g., while they indicate the number of mass extinctions, these are not Type 1 or 2 mass extinctions but simply the largest that stand out in box plots of extinction intensities [Kocsis, pers. comm.]). Nonetheless, they analyzed three diversity curves, the raw data in the PDBD, then two corrected curves, one with classical rarefaction and the other with SQS, all three being analyzed with the four extinction metrics, and then with the ~10 million timescale and the standard geologic stages (Figure 3, but they do not indicate which curve goes with which analysis). Their analysis is fully congruent with the results from the earlier analysis of the PBDB (Alroy, 2008), recognizing the Big 5 as Type 4 mass extinctions.

### A robust picture of Phanerozoic extinction intensities? Remarkable congruence between the analyses of Sepkoski's data and PBDB

The two approaches for compensating for the incompleteness of the fossil record, Foote's (2003) analysis of Sepkoski's genus compendium and Kocsis et al.'s (2019) of the PBDB, are very different, yet they yield strikingly similar results (Figure 3). Here I only refer to Kocsis et al.'s (2019) stage-level analyses of the PBDB, not their coarser 10-million year interval analyses. The congruence between Foote's (2003) and Kocsis et al.'s (2019) analyses suggests we have a robust understanding of the Phanerozoic marine extinction rates at the resolution of the International

Geologic Timescale (although it is also possible that both analyses suffer from some as yet uncompensated bias in the fossil record). Five similarities stand out.

### Zero extinction rates

Unlike the uncorrected Sepkoski and PBDB data, both studies identify multiple stages with zero (or near zero) extinction rates. Both analyses agree on the presence of a quiescent stage in the Ordovician, at least one in the Early Permian, one in the Early Jurassic, one or two stages in the Cretaceous, and two clusters of stages in the Paleogene. Foote (2003) also found a mid-Devonian, and several other Jurassic stages with rates indistinguishable from zero. The fact that the two very different analyses find highly congruent and the unanticipated pattern of zero extinction suggests that both have accurately captured the true underlying signal in the marine fossil record. This pattern has major macro-evolutionary implications, for it indicates that the marine biosphere may have extended periods of 5+ million years of remarkable ecological and evolutionary stasis, warranting further attention.

### Confirmation (and sharper extinction peaks) of the Big Five

Both analyses lead to more pronounced extinction peaks, suggesting that the Signor–Lipps effect smears extinction patterns back one to two stages in the raw data prior to the actual extinction interval (Foote, 2003; Lu et al., 2006). The peaks identified by Raup and Sepkoski (1982) are also the biggest in both analyses in the post-early Ordovician; both agree on the Biggest Five and that they are Type 4 mass extinctions.

### Identification of the same lesser peaks

Both analyses identify two peaks in the Carboniferous, an end-Jurassic peak, a mid-Cretaceous peak, and end-Eocene peak, and Pliocene peak or peaks. Kocsis et al. (2019) also find a small Capitanian peak.

### Verification of sustained Devonian extinction(s)

In Sepkoski's genus-level data, three of the last four stages of the Devonian show elevated extinction rates (the Late Devonian's Famennian, Frasnian, and the Mid-Devonian Eifelian), while in Foote's (2003) analysis there are two peaks of elevated extinction, the last two stage of the Devonian (Famennian, Frasnian). The Kocsis et al. (2019) analysis shows a similar pattern, but with elevated rates in the Mid-Devonian as well, as does Hoyal Cuthill et al. (2020), closer to the uncorrected Sepkoski data.

Thus, it appears that while there is a late-Devonian peak (Figure 3), the Devonian extinctions also represent a time of long-term extinction, first seen in Raup and Sepkoski's (1982) data, rather than representing a smearing back of a shorter-term event. The fact that the corrected data support an extended time of crisis adds support to the hypothesis that the Devonian extinctions may be due to the transition from a nonforested to forested world (Algeo et al., 1995; Algeo, 1998; Algeo and Scheckler, 2010; Lu et al., 2021), rather than being a simple perturbation that the biosphere then recovered from. The hypothesis of a long-term change in the state also finds support in the ecological analyses of (McGhee et al., 2004; McGhee et al., 2013; Muscente et al., 2018) (see below).

### Reduced origination rates for the late Devonian and end-Triassic likely an artifact

In contrast to Bambach et al. (2004) who found decreased origination rates for the late Devonian and end-Triassic, both Foote (2003) and Kocsis et al. (2019) found increasing origination rates from the middle to the end of both the Devonian and Triassic, suggesting that the end-Devonian and end-Triassic diversity losses were sharper than implied by Bambach et al.'s (2004) analysis. This does not undermine the notion that depressed origination can drive diversity loss (Quental and Marshall, 2013), but it suggests the distinction between a mass extinction and a mass depletion is not required when discussing the Big Five.

### Other measures of significance

### Ecological impact

One of the primary interests in mass extinctions is the role they played in resetting the ecological stage, paving the way for evolutionary innovation (e.g., see Jablonski, 2001; Bush et al., 2020; Hoyal Cuthill et al., 2020). Key questions for each extinction include: the degree of higher taxonomic turnover (a proxy for a major ecological change) (e.g., see Muscente et al., 2018); the degree of functional diversity change (e.g., Dunhill et al., 2018b); the selectivity on different life modes (e.g., Jablonski and Raup, 1995; Kiessling et al., 2007; Dunhill et al., 2018a, 2018b); whether geographic range buffered against extinction (e.g., Raup and Jablonski, 1993; Payne and Finnegan, 2007); the extent of biogeographic change (Kocsis et al., 2018); and so forth. A comprehensive review of this literature is beyond the scope of this paper, but it has long been clear that the correlation between extinction magnitude and ecological change depends on the extinction. For example, McGhee et al. (2013) showed using Droser et al.'s (2000) 4-level qualitative scheme for assessing the ecological impact that while most of the Big Five had an appreciable ecological impact, the end-Ordovician had almost no long-term ecological consequences.

### Ecological change is not just driven by extinction

As emphasized by others (e.g., Muscente et al., 2018; Bush et al., 2020; Hoyal Cuthill et al., 2020) ecological change is not just driven by mass extinctions, but also occurred during times of expansion of the biosphere. These times include the initial expansion of the biosphere during the Cambrian and Great Ordovician Biodiversification Event (Muscente et al., 2018; Hoyal Cuthill et al., 2020).

### Summary: The Big Five mass extinctions correspond to the biggest 5 after the early Ordovician and grade into several other major events

At the coarse temporal resolution of the geological stages in the marine realm the Big Five correspond to the biggest five extinction intervals after the early Ordovician, but also form part of a continuum of extinction intensities, with another six or so time intervals that could also be regarded as mass extinctions, including two peaks in the Carboniferous, an end-Jurassic peak, a mid-Cretaceous peak, an end-Eocene peak, a Pliocene peak or peaks, and possibly a small Capitanian peak. In the two analyses that correct for the incompleteness of the fossil record, the end-Permian is always the most severe, followed by the end-Triassic, in contrast to Raup and

Sepkoski's (1982) initial analysis at the family level where the end-Triassic was the least intense of the Big Five.

In all analyses, the end-Permian stands out, so if you wanted to be conservative, maybe there was only the Big One (Alroy, 2008), but you could make the case for the Big Two, the Big Three, all the way down to about the Big Dozen. In terms of the ecological and biogeographic impact, the end-Permian is always ranked the highest followed by the end-Triassic or end-Cretaceous, then the Late Devonian. In the ecological analyses, the end-Ordovician is at most a minor extinction event.

## Is the sixth mass extinction a mass extinction? Not yet, but it could be

### Untangling rate from magnitude and the status of the sixth mass extinction

Analysis of the current biodiversity crisis has led to the discussion of the relationship between elevated rates of extinction versus the magnitude of extinction, for example, see Barnosky et al. (2011). The difficulty is how to compare the present with the fossil record, given the enormity of geologic time.

### Rate

In terms of rate, it appears that the current loss of species may be the fastest ever experienced by the biosphere (Barnosky et al., 2011), except for the end-Cretaceous. The end-Cretaceous is likely to have been even faster than today's losses, especially if the driver of the extinctions was rapid bolide-induced climate change (Chiarenza et al., 2020), including the blocking of sunlight due to the impact debris with a resulting decrease in photosynthesis for a few years (Bardeen et al., 2017; Tabor et al., 2020), and/or due to cooling of 26 °C or more with 3–16 years of freezing temperatures (Brugger et al., 2017). The Cambrian biomere extinctions (Palmer, 1984, 1998; Saltzman et al., 2015) may also have been very rapid, depending on the timescale over which anoxic waters flooded the shallow water shelves.

### Magnitude

But in terms of magnitude, the loss of species to date compared with the species loss in the fossil record is very small. So, does the current biodiversity crisis count as a mass extinction? In the human realm, an analogy might be the loss of life in New York at the World Trade Center on 9/11. In terms of casualties per hour, the rate rivals the highest casualty rates of many of the worst battles in human history, swamping the rates of loss of life in the great plagues. But in terms of magnitude, the loss of life on 9/11 was small compared with these other disasters. Nonetheless, it was still unprecedented in the historical context in which it happened, as is the current biodiversity crisis.

### Future losses

A key point is not what has happened to date in the current biosphere but what may happen (Ceballos et al., 2020). If the rate of loss of life that occurred on 9/11 persisted for many hours, days, weeks, or months, the losses would have been truly monumental. Similarly, if the current rate of species loss continues (or accelerates) for centuries, millennia, or tens of millennia (still only 0.01 million years), then the current crisis will truly be the beginning of a mass extinction. But so far, the absolute magnitude of extinction is still very small compared with the Big Five; there is hope yet that intervention can prevent us going too far down that path.

### What to call the sixth mass extinction?

When paleontologists talk of the Big Five, they usually have in mind the specific events identified by Raup and Sepkoski in 1982. But, had the first analysis of the Phanerozoic marine realm been with the PBDB, coinformed with data on ecological impact, it is unclear how many mass extinctions would have been identified. The end-Permian, end-Cretaceous, and end-Triassic always stand out, but so also do the Cambrian events (at least in terms of per lineage intensity). Moreover, the temporal resolution is so coarse in the rock record that it is very difficult to compare rates in the past with rates in the present, so it is unclear how to rank the current crisis with the many events we see in the fossil record. So, what, exactly would we have called the current biodiversity crisis? Not the Sixth Mass Extinction I would warrant. Perhaps the Incipient Mass Extinction, or the Incipient Anthropocene Mass Extinction, would be appropriate terms, with the deeply disturbing corollary that it might come to rank among the Big Five.

## Future directions

### Back to the outcrop: The rocky road ahead

Ultimately, the time interval over which taxa went extinct, and the environmental conditions that correlate with those extinctions, can only be gleaned from outcrop-level analyses. However, this type of analysis is not simple to undertake.

### The highly structured (and thus hard to read) rock record

The fact that sedimentary rock is highly structured temporally and spatially, and that marine biodiversity is structured by water depth, complicates the interpretation of the observed patterns of first and last occurrences in the fossil record (Holland, 1995), including the interpretation of mass extinctions (Holland and Patzkowsky, 2015). The recognition that sea-level rise and fall might compromise the ability to read the fossil record has long been appreciated (e.g., see Newell, 1952; Hallam and Wignall, 1999; Smith and McGowan, 2007; McGowan and Smith, 2008; Peters, 2008).

The structure of the rock record is now relatively well understood, falling under the rubric of sequence stratigraphy (Holland, 2020). There are two important points. First, the representation of a given environment in the sedimentary rock record is patchy and typically bounded by temporal hiatuses both above and below. This can produce concentrations of last occurrences that correspond to the disappearance of the appropriate environment and not necessarily to the actual time of extinction of the taxa. Holland (2020) notes that all the mass extinctions, except for the end-Cretaceous, have last occurrences clustered at sequence stratigraphic horizons (at maximum flooding surfaces, whether accompanied by subaerial exposure or not, and at surfaces of forced regression [SFRs]) which means that the rock record may well be superimposing a false appearance of pulsed extinctions. In fact, in some cases, we expect single extinction events to be expressed as double peaks in the rock record (Holland and Patzkowsky, 2015), which has certainly complicated the interpretation of the end-Ordovician extinction (Zimmt et al., 2021). Recently, Zimmt et al. (2021) developed an approach for determining whether clusters of last occurrences

represent real extinctions or are just artifacts of the rock record. We (Zimmt et al., 2021) propose only analyzing taxa for which the appropriate environment (facies) is represented above the putative extinction interval. This approach has yet to be applied to field data.

The second point, flowing directly from the first, is that the duration of the extinction interval is likely to be longer than a direct reading of the fossil record in local sections would indicate (except for the end-Cretaceous). How much longer has yet to be determined, but Holland (2020) indicates it could be tens to hundreds of thousands of years or longer. It seems unlikely that most mass extinctions can be dated to precisions higher than the durations of the hiatuses, the temporal gaps, between the packages of sedimentary rocks that carry the fossil record, despite the impressive increase in the precision of radiometric dating. Among the Big Five, the only exception is the end-Cretaceous, where the extinctions do not correspond to hiatal surfaces. In this case, the increased precision of radiometric dates is particularly valuable, for example, the end-Cretaceous is now dated to 66.043 million years $\pm0.043$ million ($2\sigma$) years (Sprain et al., 2018).

## Spatial data

A second area that has yet to be explored properly is the spatial structure of biodiversity change. Most analyses of the patterns of Phanerozoic extinction have treated the data globally. However, Close et al. (2020) recently analyzed regional Phanerozoic diversity trajectories taking into account the spatial inhomogeneity in the rock record and the environments they represent, as did Miller (1997) for the Ordovician radiation. It will be very interesting to see how a similar analysis of extinction rates might alter our current understanding.

## Understanding the mechanisms of extinction

When Raup and Sepkoski (1982) first proposed the Big Five mass extinctions we knew essentially nothing about their causes. Forty years later we have come a long way in understanding the causes of some of the largest events, the roles that big extinctions play in macroevolutionary processes, have a much better knowledge of the fossil record, and an ever-growing set of tools for measuring environmental conditions from the rock record. With the development of Earth Systems models, for example, the Community Earth System Model (CESM: https://www.cesm.ucar.edu/), which models mass, energy, and momentum exchanges between the atmosphere, ocean, and land along with biogeochemical cycles of $O_2$, carbon, and nutrients, we are now poised to understand how extinctions result from interactions between the biosphere and the physical and chemical components of the Earth System (Penn et al., 2018; Dal Corso et al., 2022; Reddin et al., 2022). My perception is that we now stand on the threshold of understanding the nature of extinction observed in the fossil record, something that was unimaginable when I was a graduate student in the mid-1980s.

**Open peer review.** To view the open peer review materials for this article, please visit http://doi.org/10.1017/ext.2022.4.

**Acknowledgements.** Thanks to Wolfgang Kiessling for the invitation to write this paper; to Carl Reddin for his especially comprehensive review; to Alex Dunhill for his review; to Barry Brook, Seth Finnegan, Michael Foote, Ádám Kocsis, Arnold Miller, Swee Peck Quek, and Haijun Song for input; and Tetsuto Miyashita for drawing my attention to the Romer quote.

**Financial support.** This work was partially supported by the Philip Sandford Boone Chair in Paleontology at the University of California, Berkeley.

**Competing interest.** The author declares none.

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
