## [Reviewer Report]

*Comments to Author*: The review by Marshall is an informative and very useful discussion of the past and future of the identification and definition of mass extinctions. Though it will be a great contribution when finished, it currently reads like a raw draft. In particular, the headings, subheadings, sub-subheadings, etc, are quite confusingly ordered and need sorting out (e.g. no more than three levels of headings should be used). Some parts may be better extracted and presented as a box separate from the main text, or bullet points. I think there is sufficient structure already there but the headings currently do more to confuse than to help. The figures could be improved with a little work, as suggested below. There are quite a number of grammatical errors in this manuscript that a proper proof-read is needed to identify. I did my best to highlight a few but did not dwell on them. I have also noted that several of the assertions herein are open to debate and should be noted as such, where this is the case. Minor comments (the L432 comment about sample standardization below may be a bit more serious) are added below with the line numbers as on the pdf I was provided.

Perhaps for historical reasons, the review focuses on analyses of Sepkoski’s compendium (e.g. in Table 1, 3 out of 4 of the extinction rankings are base on Sepkoski’s compendium) and only secondarily covers analyses of the PBDB, despite highlighting many of the benefits of using the PBDB. Historical reasons would be fine (they should be made explicit, if so), but it currently reads as if the Sepkoski’s compendium analyses are preferred for inference on matters of mass extinctions, but I suspect the author did not intend this? The author could also amend this by writing more openly about advantages that either the PBDB or Sepkoski’s compendium have for analysis, and clearer structuring to the headings and subheadings (e.g. if they are partly following a historical timeline).

Figures.

Figure 3 is a bit odd and is only needed for two of the five factors. The other three are just listed. I am not sure of the numerical support underlying this figure. Please be clear whether this is just a schematic or whether it is a real example with data underlying it.

Definition of “correcting for the incompleteness of the fossil record”? Some note should be made that this is an attempt at correction, but many biases no doubt remain.

For temporal resolution, perhaps the ~10-myr bin some studies use should be mentioned? E.g. Alroy et al. 2008

Figure 4. Comparison of these two panels is not that easy, mainly because of the distance between them and the different presentations of the two panels. Consider pushing the two panels together to aid comparison for the reader. They are also a bit fuzzy – the raw pdf version of this panel is available at https://github.com/divDyn/ddPhanero/blob/master/export/0.4/sumRates.pdf Consider extracting the data points and error bars from the Foote figure using data extraction software (e.g. the simple DataThief software) and replotting with new axes and style that is consistent with the Kocsis et al. 2019 figure.

Abstract.

L33. Second quotation mark

L37. Perhaps, “are not always statistically distinct”. Also, in all cases, distinct from what? Needs adding that we are talking about distinction from background rates

Keywords: compendium

Impact Statement.

L16. Add quotation marks around “the sixth mass extinction”

L22 Perhaps add quotation marks to “the Big 3” on first mentioning

L23. LIKELY extended over hundreds of thousands of years. We will probably never know how fast many of these events were, as our understanding is limited by the stratigraphic resolution, as mentioned later (Holland 2020).

Introduction.

L81. Perhaps “discuss” rather than “answer”, since I think some of these questions will rightfully remain open.

L82. Delete “the” in “the outstanding questions” as I’m sure there will be others not listed.

L95. “The relationship between the nature of the rock record and the fossil record that it carries”. Perhaps add Peters 2008 in here: Peters, S. E. (2008). Environmental determinants of extinction selectivity in the fossil record. Nature, 454(7204), 626-629.

L118. “that there were two events in the Permian”. Perhaps add a reference supporting here, or add note to later analysis

L133. I think you should specify if you mean genus or family extinction rates in “high extinction rates” (I assume family, given the subheading)

L138. Should have been

L158-9. Delete “the” in “the genus-level data” and “the family-level data”. Maybe a justification of why genus data is now standard would be useful here.

L168: “concluded” seems a too strong word for this statement. “remarked”?

L173: “the genus level data”. Is this specifically referring to Sepkoski’s genus level data or all genus level fossil data e.g. in PBDB too? Perhaps the points made here are of relevance wider than just Sepkoski’s data

L190: a reminder of the definition of paraphyletic would be useful here

L194: “extinction of genus name”?

L224-5: “Stanley (2016) estimates the corresponding species-level extinction to be 81%,”. This is probably worth noting that it may be a better answer than Raup’s?

Two Meanings of the Term Mass Extinction. Perhaps a sentence or two justifying the importance of identifying a set number of mass extinctions would be useful? Why should we care if there are three, six or 18?

L244-5. Use “extinction” or “mass extinction” but not both here

L256. Paleobiology Database needs link to website adding

L258. Why is the fourth factor not given its own paragraph like the preceding three? Perhaps bullet points would be better for each.

L265 and rest. Consider putting section “Some Mass Extinction Terminology” in a separate box as it seems separate to the rest of the storyline

L290-1. I think this mathematical notation for temporal scope is unnecessary and distracting

L315. Not clear what the hyphen is doing. I think it is intended as punctuation, in which case replace with a period/full-stop.

L318. Also see L526 later. Passage on “Changing the Temporal Resolution Adds Another Big Extinction”. It looks arguable whether the Capitanian would also be classed as a “big” extinction e.g. including in Kocsis et al. 2019, the Capitanian stage-level extinction rates are not particularly high even in the maximum rate given across the different analytical approaches. In my experience the Capitanian only really emerges as substantial in extinction rate from the Sepkoski’s compendium, and rarely if ever from the PBDB (e.g. also in Hoyall-Cuthill et al. 2020). Why might this extinction be exclusive to Sepkoski’s compendium would be worth discussing. For example, as discussed later, the dearth of true occurrences of taxa in Sepkoski’s compendium could mean that it is mathematically difficult to compare sampling rates among intervals, such that intervals of low and high sampling completeness are poorly contrasted. I suspect the Capitanian, though may be a true event, is artificially enlarged by its proximity to the PT, which is particularly apparent when data are not standardised for sampling completeness. McGhee et al. 2013 add: “the Capitanian: it drops from third-most severe in Table 1 to ninth-most severe in the eleven traditional biodiversity crises in Table 3 (in actuality, the Capitanian is ranked still lower at 32 in a total of 66 mid-Ordovician through Cretaceous stages). This radical difference is largely due to updated age assignments in the Paleobiology Database, reflecting improvements to the Permian timescale in the last decade.”

L332. Check Fan et al. 2020 for their higher temporal resolution to the Triassic. Fan, J. X., Shen, S. Z., Erwin, D. H., Sadler, P. M., MacLeod, N., Cheng, Q. M., ... & Zhao, Y. Y. (2020). A high-resolution summary of Cambrian to Early Triassic marine invertebrate biodiversity. Science, 367(6475), 272-277.

L350: “would make it difficult to justify excluding the Cambrian”. You mean also if the temporal resolution and confidence were higher, which they are not, which I believe are often reasons for excluding it.

L355. Fan et al 2020 should be cited in here

L358. Paragraph on “The Temporal Resolution Can Be Too High”. Can they really be “too high” or does a higher resolution just emphasise different events? Perhaps a distinction should be made between extinction events that are drawn out over 10s of million years, like the Late Devonian (unless a higher resolution clarifies them into several rapid events), and events that are much more rapid, like the KT and PT. Both may be extinction events, but the latter events may be more interesting as case studies for the future of the modern extinction crisis, while drawn out extinctions likely have less applicable to the modern, but may be interesting for macroevolution.

L374. Geographical completeness deserves a mention here e.g. Close et al. 2020, that different times are represented by different paleogeographical foci of the according rock record, biasing the end patterns observed. So it is not just the extent of fossiliferous rock per interval but the paleogeographical representation of that rock. Additional linked factors are biases of variation in lithological and environmental, e.g. depth, representations over different time intervals, and the sampling completeness of different taxa e.g. most echinoderm groups are more poorly sampled than e.g. scleractinian corals.

Close, R. A., Benson, R. B., Saupe, E. E., Clapham, M. E., & Butler, R. J. (2020). The spatial structure of Phanerozoic marine animal diversity. Science, 368(6489), 420-424.

L378. “although this aspect of the incompleteness of the fossil record is unlikely to create spurious extinction peaks”. But it could, albeit peaks that are likely lesser than the true event peak. But if a large extinction event was preceded by a low completeness interval, and that interval preceded by a high completeness interval, LADs would be expected to cluster artificially in the high completeness interval (despite truly going extinct at the large extinction event).

L393. “presence of three extinction regimes”. Perhaps this should be introduced with a note of caution since even the ‘mass extinctions’ (however defined) may simply represent the end of a continuum of intervals ranked by extinction rates. The ‘zero rate’ intervals could possibly just represent a mathematical curiosity derived from the incompleteness of the fossil record, without any real macroevolutionary meaning. ‘regimes’ to me notes some real qualitative difference, which should be confirmed first before claiming the presence of regimes.

L432. “sometimes observed differences in species (or genus) occurrences might be real and not need to be standardized (Bush et al., 2004).” This appears to be simply an opinion without real quantitative appraisal. Note in the example given, as mentioned, one still finds the correct answer that the Arctic is more species-poor. Naturally, sampling standardisation tosses out valid occurrences and species presences, but the point of standardisation is to make two samples more comparable in a relative sense. Not to make either species list more complete in an absolute sense. This underlies all rarefaction approaches and species accumulation curves. Afterall, no species inventory is truly saturated (species will always be missed) and all are dependent on sampling intensity. Thus we must compare relative richnesses rather than absolute richnesses. Absolute richnesses are interesting in their own right, e.g. species membership, but not in comparing two or more sites/times.

“We don’t know the extent to which sample standardization distorts the underlying diversity (and thus extinction rate) signal”. With this sentence the author seems to believe sampling standardisation universally distorts a diversity signal? I hope that is not the intended meaning here as I would strongly disagree. As above, we can only achieve a signal of relative differences in diversity, even with modern species censuses, and sampling standardisation (ideally both a priori, e.g. deciding on sampling 1 m^3 rock volume with similar lithology and sedimentation rates from two sites, and a posteriori e.g. rarefaction) allows that relative count to be compared with less bias.

L441. We know it’s not true for either datasets e.g. Adrain and Westrop, 2000, as mentioned earlier, though Sepkoski’s compendium will be more taxonomically harmonised than the PBDB. Also, what does “taxonomically complete” mean here? Surely taxonomy is a relatively dynamic field that is never complete? I think the author means primarily avoiding synonymous species names. I would doubt that genera in Sepkoski’s compendium can be compared with those in the PBDB simply because time has moved on and taxa now have different optimal naming and taxonomic structuring.

L445. This sentence is incomplete. “representative,”… of what? Representative of true diversity patterns?

L466. “that the Signor-Lipps effect smears extinction patterns back 1 to 2 stages”… I think “in the raw data” is missing.

L492. “episodes were sharper”. In what way? Episodes of sharper high origination rates or sharper diversity loss?

L512. Consider adding Bush et al. 2020 in here discussing effects of extinction magnitude and selectivity? Bush, A. M., Wang, S. C., Payne, J. L., & Heim, N. A. (2020). A framework for the integrated analysis of the magnitude, selectivity, and biotic effects of extinction and origination. Paleobiology, 46(1), 1-22.

L517. “produced” rather than “had”

L519 and 529. These two subtitles seem to be almost duplicated. The first paragraph seems unrelated to its title: “In a Qualitative Analysis the End-Ordovician Falls Out of the Biggest Five” and “In a Quantitative Analysis the End-Ordovician Also Falls Out of the Biggest Five”

L534. “but by the end-Cretaceous and most of the Devonian and into the early Carboniferous at the family and genus levels”. Sentence seems incomplete…

L542. Just to check, the following statement refers to the results of Muscente et al 2018? “at the highest level ecological innovation is decoupled from large extinction events”

L568: “minor EXTINCTION event”

L572. Consider adding Neubauer, T. A., Hauffe, T., Silvestro, D., Schauer, J., Kadolsky, D., Wesselingh, F. P., ... & Wilke, T. (2021). Current extinction rate in European freshwater gastropods greatly exceeds that of the late Cretaceous mass extinction. Communications Earth & Environment, 2(1), 1-7.

L577. I think “small number of species losses to date” needs further qualification here, since some groups e.g. large terrestrial mammals are subject to high species losses already. Perhaps something like “small relative to species losses at ancient extinction events”

L581. Add a reference for discussion of rates of Cambrian biomere extinctions? Also, add associated rapid global cooling to “blocking of sunlight”.

L590. This is an excellent analogy.

L605: “not simply undertaken”. You mean “not simple to undertake”?

L612: add Peters 2008 to this list? Also lines 618 to 622

L623. “to be expressed”

L645: Unclear what is meant by “Earth Systems approaches”. Earth Systems modelling? Consider adding “mechanism-based simulations” as a useful modern tool e.g. as used in Penn et al. 2018 or Reddin et al. 2022.

Penn, J. L., Deutsch, C., Payne, J. L., & Sperling, E. A. (2018). Temperature-dependent hypoxia explains biogeography and severity of end-Permian marine mass extinction. Science, 362(6419), eaat1327.

Reddin, C. J., Aberhan, M., Raja, N. B., & Kocsis, Á. T. (2022). Global warming generates predictable extinctions of warm-and cold-water marine benthic invertebrates via thermal habitat loss. Global Change Biology.

---

## [Reviewer Report]

*Comments to Author*: This is a nice summary of the current knowledge we have on mass extinctions that thoughtfully discusses and interprets what we mean by mass extinction. I’d be happy to see it published with a few considerations based on the comments below. I would certainly add it to the reading list for my UG extinction lecture/practical.

Lines 140-142: Does a single sentence needs its own sub-heading? Looks a bit weird to me but that might just be my own preference. I’d also add that this is a major point for mass extinctions i.e. they cause a prolonged drop in standing global diversity (i.e. gamma diversity) that is recognisable in the fossil record.

Lines 149-: Some mass extinction workers argue for magnitude calculations rather than rates - as extinction rate calculations might underestimate as presume extinction extended across entire stages e.g. the Rhaetian is ~ 4Myrs long but the ETE lasted around 30Kyrs - a rate calculation assumes the extinction occurred across all 4 million years. See Dunhill et al. 2018AB and Kiessling et al. 2007 in ref list below.

Lines 168-171: I agree with Raup’s statement but I believe it lies in selectivity differences i.e. ecological processes - the difference between the field of bullets or stressor-specific extinction. Dunhill et al. 2018A show distinctly different extinction selectivity between ETE and the background intervals either side despite those 'background' intervals also showing relatively high extinction rates compared to background levels throughout the Phanerozoic. Also, note the geographic range selectivity between background and mass extinction – i.e. Payne and Finnegan 2007.

Lines 206-: RE the Cambrian - I would add that this is where drop in standing diversity becomes important - would it be fair to say that we don't recognise any major mass extinctions in the Cambrian because those high extinction rates are outstripped by even higher origination rates? That said, you get bursts of high extinction but these are accompanied by bursts of even higher origination so you're getting turnover events rather than mass extinction events, which often also go hand in hand with low origination rates (until the recovery kicks in, that is).

Lines 512-: I’d come back to selectivity here again, i.e. that a change in ecological selectivity can be a signal of mass extinction - either a switch to generalistic extinction where selectivity becomes less strong (i.e. with regard to geog range) or stressor-specific extinction signals appear (i.e. reef taxa, benthic taxa, apex predators etc.) – see Payne and Finnegan 2007, Dunhill et al. 2018A, Pimiento et al. 2017,

Cheers,

Alex Dunhill

University of Leeds

References:

Dunhill, A. M., W. J. Foster, S. Azaele, J. Sciberras and R. J. Twitchett (2018A). "Modelling determinants of extinction across two Mesozoic hyperthermal events." Proceedings of the Royal Society B: Biological Sciences 285(1889).

Dunhill, A. M., W. J. Foster, J. Sciberras and R. J. Twitchett (2018B). "Impact of the Late Triassic mass extinction on functional diversity and composition of marine ecosystems." Palaeontology 61(1): 133-148.

Kiessling, W., M. Aberhan, B. Brenneis and P. J. Wagner (2007). "Extinction trajectories of benthic organisms across the Triassic–Jurassic boundary." Palaeogeography, Palaeoclimatology, Palaeoecology 244(1–4): 201-222.

Payne, J. L. and S. Finnegan (2007). "The effect of geographic range on extinction risk during background and mass extinction." Proceedings of the National Academy of Sciences of the United States of America 104(25): 10506-10511.

Pimiento, C., J. N. Griffin, C. F. Clements, D. Silvestro, S. Varela, M. D. Uhen and C. Jaramillo (2017). "The Pliocene marine megafauna extinction and its impact on functional diversity." Nature Ecology & Evolution 1(8): 1100-1106.

---

## [Editor Report]

*Comments to Author*: Dear Prof. Marshall,

Thank you for submitting your manuscript to Cambridge Prisms: Extinction. I have now received reports from two reviewers. After careful consideration, I have decided to invite a minor revision of the manuscript. Collectively, all reviewers agree that this paper is interesting and significant and should be published in Cambridge Prisms: Extinction. As you will see from the reports, the reviewers raise numerous concerns in the text and some figures. If you feel that you can comprehensively address the reviewers’ concerns, please provide a point-by-point response to these comments along with your revision. If you cannot address specific reviewer requests or find any points invalid, please explain why in the point-by-point response.

Sincerely,

Haijun

Prof. Haijun Song

Handling Editor, Cambridge Prisms: Extinction

Email: haijunsong@cug.edu.cn 

Reviewer 1

The review by Marshall is an informative and very useful discussion of the past and future of the identification and definition of mass extinctions. Though it will be a great contribution when finished, it currently reads like a raw draft. In particular, the headings, subheadings, sub-subheadings, etc, are quite confusingly ordered and need sorting out (e.g. no more than three levels of headings should be used). Some parts may be better extracted and presented as a box separate from the main text, or bullet points. I think there is sufficient structure already there but the headings currently do more to confuse than to help. The figures could be improved with a little work, as suggested below. There are quite a number of grammatical errors in this manuscript that a proper proof-read is needed to identify. I did my best to highlight a few but did not dwell on them. I have also noted that several of the assertions herein are open to debate and should be noted as such, where this is the case. Minor comments (the L432 comment about sample standardization below may be a bit more serious) are added below with the line numbers as on the pdf I was provided.

Perhaps for historical reasons, the review focuses on analyses of Sepkoski’s compendium (e.g. in Table 1, 3 out of 4 of the extinction rankings are base on Sepkoski’s compendium) and only secondarily covers analyses of the PBDB, despite highlighting many of the benefits of using the PBDB. Historical reasons would be fine (they should be made explicit, if so), but it currently reads as if the Sepkoski’s compendium analyses are preferred for inference on matters of mass extinctions, but I suspect the author did not intend this? The author could also amend this by writing more openly about advantages that either the PBDB or Sepkoski’s compendium have for analysis, and clearer structuring to the headings and subheadings (e.g. if they are partly following a historical timeline).

Figures.

Figure 3 is a bit odd and is only needed for two of the five factors. The other three are just listed. I am not sure of the numerical support underlying this figure. Please be clear whether this is just a schematic or whether it is a real example with data underlying it.

Definition of “correcting for the incompleteness of the fossil record”? Some note should be made that this is an attempt at correction, but many biases no doubt remain.

For temporal resolution, perhaps the ~10-myr bin some studies use should be mentioned? E.g. Alroy et al. 2008

Figure 4. Comparison of these two panels is not that easy, mainly because of the distance between them and the different presentations of the two panels. Consider pushing the two panels together to aid comparison for the reader. They are also a bit fuzzy – the raw pdf version of this panel is available at https://github.com/divDyn/ddPhanero/blob/master/export/0.4/sumRates.pdf Consider extracting the data points and error bars from the Foote figure using data extraction software (e.g. the simple DataThief software) and replotting with new axes and style that is consistent with the Kocsis et al. 2019 figure.

Abstract.

L33. Second quotation mark

L37. Perhaps, “are not always statistically distinct”. Also, in all cases, distinct from what? Needs adding that we are talking about distinction from background rates

Keywords: compendium

Impact Statement.

L16. Add quotation marks around “the sixth mass extinction”

L22 Perhaps add quotation marks to “the Big 3” on first mentioning

L23. LIKELY extended over hundreds of thousands of years. We will probably never know how fast many of these events were, as our understanding is limited by the stratigraphic resolution, as mentioned later (Holland 2020).

Introduction.

L81. Perhaps “discuss” rather than “answer”, since I think some of these questions will rightfully remain open.

L82. Delete “the” in “the outstanding questions” as I’m sure there will be others not listed.

L95. “The relationship between the nature of the rock record and the fossil record that it carries”. Perhaps add Peters 2008 in here: Peters, S. E. (2008). Environmental determinants of extinction selectivity in the fossil record. Nature, 454(7204), 626-629.

L118. “that there were two events in the Permian”. Perhaps add a reference supporting here, or add note to later analysis

L133. I think you should specify if you mean genus or family extinction rates in “high extinction rates” (I assume family, given the subheading)

L138. Should have been

L158-9. Delete “the” in “the genus-level data” and “the family-level data”. Maybe a justification of why genus data is now standard would be useful here.

L168: “concluded” seems a too strong word for this statement. “remarked”?

L173: “the genus level data”. Is this specifically referring to Sepkoski’s genus level data or all genus level fossil data e.g. in PBDB too? Perhaps the points made here are of relevance wider than just Sepkoski’s data

L190: a reminder of the definition of paraphyletic would be useful here

L194: “extinction of genus name”?

L224-5: “Stanley (2016) estimates the corresponding species-level extinction to be 81%,”. This is probably worth noting that it may be a better answer than Raup’s?

Two Meanings of the Term Mass Extinction. Perhaps a sentence or two justifying the importance of identifying a set number of mass extinctions would be useful? Why should we care if there are three, six or 18?

L244-5. Use “extinction” or “mass extinction” but not both here

L256. Paleobiology Database needs link to website adding

L258. Why is the fourth factor not given its own paragraph like the preceding three? Perhaps bullet points would be better for each.

L265 and rest. Consider putting section “Some Mass Extinction Terminology” in a separate box as it seems separate to the rest of the storyline

L290-1. I think this mathematical notation for temporal scope is unnecessary and distracting

L315. Not clear what the hyphen is doing. I think it is intended as punctuation, in which case replace with a period/full-stop.

L318. Also see L526 later. Passage on “Changing the Temporal Resolution Adds Another Big Extinction”. It looks arguable whether the Capitanian would also be classed as a “big” extinction e.g. including in Kocsis et al. 2019, the Capitanian stage-level extinction rates are not particularly high even in the maximum rate given across the different analytical approaches. In my experience the Capitanian only really emerges as substantial in extinction rate from the Sepkoski’s compendium, and rarely if ever from the PBDB (e.g. also in Hoyall-Cuthill et al. 2020). Why might this extinction be exclusive to Sepkoski’s compendium would be worth discussing. For example, as discussed later, the dearth of true occurrences of taxa in Sepkoski’s compendium could mean that it is mathematically difficult to compare sampling rates among intervals, such that intervals of low and high sampling completeness are poorly contrasted. I suspect the Capitanian, though may be a true event, is artificially enlarged by its proximity to the PT, which is particularly apparent when data are not standardised for sampling completeness. McGhee et al. 2013 add: “the Capitanian: it drops from third-most severe in Table 1 to ninth-most severe in the eleven traditional biodiversity crises in Table 3 (in actuality, the Capitanian is ranked still lower at 32 in a total of 66 mid-Ordovician through Cretaceous stages). This radical difference is largely due to updated age assignments in the Paleobiology Database, reflecting improvements to the Permian timescale in the last decade.”

L332. Check Fan et al. 2020 for their higher temporal resolution to the Triassic. Fan, J. X., Shen, S. Z., Erwin, D. H., Sadler, P. M., MacLeod, N., Cheng, Q. M., ... & Zhao, Y. Y. (2020). A high-resolution summary of Cambrian to Early Triassic marine invertebrate biodiversity. Science, 367(6475), 272-277.

L350: “would make it difficult to justify excluding the Cambrian”. You mean also if the temporal resolution and confidence were higher, which they are not, which I believe are often reasons for excluding it.

L355. Fan et al 2020 should be cited in here

L358. Paragraph on “The Temporal Resolution Can Be Too High”. Can they really be “too high” or does a higher resolution just emphasise different events? Perhaps a distinction should be made between extinction events that are drawn out over 10s of million years, like the Late Devonian (unless a higher resolution clarifies them into several rapid events), and events that are much more rapid, like the KT and PT. Both may be extinction events, but the latter events may be more interesting as case studies for the future of the modern extinction crisis, while drawn out extinctions likely have less applicable to the modern, but may be interesting for macroevolution.

L374. Geographical completeness deserves a mention here e.g. Close et al. 2020, that different times are represented by different paleogeographical foci of the according rock record, biasing the end patterns observed. So it is not just the extent of fossiliferous rock per interval but the paleogeographical representation of that rock. Additional linked factors are biases of variation in lithological and environmental, e.g. depth, representations over different time intervals, and the sampling completeness of different taxa e.g. most echinoderm groups are more poorly sampled than e.g. scleractinian corals.

Close, R. A., Benson, R. B., Saupe, E. E., Clapham, M. E., & Butler, R. J. (2020). The spatial structure of Phanerozoic marine animal diversity. Science, 368(6489), 420-424.

L378. “although this aspect of the incompleteness of the fossil record is unlikely to create spurious extinction peaks”. But it could, albeit peaks that are likely lesser than the true event peak. But if a large extinction event was preceded by a low completeness interval, and that interval preceded by a high completeness interval, LADs would be expected to cluster artificially in the high completeness interval (despite truly going extinct at the large extinction event).

L393. “presence of three extinction regimes”. Perhaps this should be introduced with a note of caution since even the ‘mass extinctions’ (however defined) may simply represent the end of a continuum of intervals ranked by extinction rates. The ‘zero rate’ intervals could possibly just represent a mathematical curiosity derived from the incompleteness of the fossil record, without any real macroevolutionary meaning. ‘regimes’ to me notes some real qualitative difference, which should be confirmed first before claiming the presence of regimes.

L432. “sometimes observed differences in species (or genus) occurrences might be real and not need to be standardized (Bush et al., 2004).” This appears to be simply an opinion without real quantitative appraisal. Note in the example given, as mentioned, one still finds the correct answer that the Arctic is more species-poor. Naturally, sampling standardisation tosses out valid occurrences and species presences, but the point of standardisation is to make two samples more comparable in a relative sense. Not to make either species list more complete in an absolute sense. This underlies all rarefaction approaches and species accumulation curves. Afterall, no species inventory is truly saturated (species will always be missed) and all are dependent on sampling intensity. Thus we must compare relative richnesses rather than absolute richnesses. Absolute richnesses are interesting in their own right, e.g. species membership, but not in comparing two or more sites/times.

“We don’t know the extent to which sample standardization distorts the underlying diversity (and thus extinction rate) signal”. With this sentence the author seems to believe sampling standardisation universally distorts a diversity signal? I hope that is not the intended meaning here as I would strongly disagree. As above, we can only achieve a signal of relative differences in diversity, even with modern species censuses, and sampling standardisation (ideally both a priori, e.g. deciding on sampling 1 m^3 rock volume with similar lithology and sedimentation rates from two sites, and a posteriori e.g. rarefaction) allows that relative count to be compared with less bias.

L441. We know it’s not true for either datasets e.g. Adrain and Westrop, 2000, as mentioned earlier, though Sepkoski’s compendium will be more taxonomically harmonised than the PBDB. Also, what does “taxonomically complete” mean here? Surely taxonomy is a relatively dynamic field that is never complete? I think the author means primarily avoiding synonymous species names. I would doubt that genera in Sepkoski’s compendium can be compared with those in the PBDB simply because time has moved on and taxa now have different optimal naming and taxonomic structuring.

L445. This sentence is incomplete. “representative,”… of what? Representative of true diversity patterns?

L466. “that the Signor-Lipps effect smears extinction patterns back 1 to 2 stages”… I think “in the raw data” is missing.

L492. “episodes were sharper”. In what way? Episodes of sharper high origination rates or sharper diversity loss?

L512. Consider adding Bush et al. 2020 in here discussing effects of extinction magnitude and selectivity? Bush, A. M., Wang, S. C., Payne, J. L., & Heim, N. A. (2020). A framework for the integrated analysis of the magnitude, selectivity, and biotic effects of extinction and origination. Paleobiology, 46(1), 1-22.

L517. “produced” rather than “had”

L519 and 529. These two subtitles seem to be almost duplicated. The first paragraph seems unrelated to its title: “In a Qualitative Analysis the End-Ordovician Falls Out of the Biggest Five” and “In a Quantitative Analysis the End-Ordovician Also Falls Out of the Biggest Five”

L534. “but by the end-Cretaceous and most of the Devonian and into the early Carboniferous at the family and genus levels”. Sentence seems incomplete…

L542. Just to check, the following statement refers to the results of Muscente et al 2018? “at the highest level ecological innovation is decoupled from large extinction events”

L568: “minor EXTINCTION event”

L572. Consider adding Neubauer, T. A., Hauffe, T., Silvestro, D., Schauer, J., Kadolsky, D., Wesselingh, F. P., ... & Wilke, T. (2021). Current extinction rate in European freshwater gastropods greatly exceeds that of the late Cretaceous mass extinction. Communications Earth & Environment, 2(1), 1-7.

L577. I think “small number of species losses to date” needs further qualification here, since some groups e.g. large terrestrial mammals are subject to high species losses already. Perhaps something like “small relative to species losses at ancient extinction events”

L581. Add a reference for discussion of rates of Cambrian biomere extinctions? Also, add associated rapid global cooling to “blocking of sunlight”.

L590. This is an excellent analogy.

L605: “not simply undertaken”. You mean “not simple to undertake”?

L612: add Peters 2008 to this list? Also lines 618 to 622

L623. “to be expressed”

L645: Unclear what is meant by “Earth Systems approaches”. Earth Systems modelling? Consider adding “mechanism-based simulations” as a useful modern tool e.g. as used in Penn et al. 2018 or Reddin et al. 2022.

Penn, J. L., Deutsch, C., Payne, J. L., & Sperling, E. A. (2018). Temperature-dependent hypoxia explains biogeography and severity of end-Permian marine mass extinction. Science, 362(6419), eaat1327.

Reddin, C. J., Aberhan, M., Raja, N. B., & Kocsis, Á. T. (2022). Global warming generates predictable extinctions of warm-and cold-water marine benthic invertebrates via thermal habitat loss. Global Change Biology. 

Reviewer 2

This is a nice summary of the current knowledge we have on mass extinctions that thoughtfully discusses and interprets what we mean by mass extinction. I’d be happy to see it published with a few considerations based on the comments below. I would certainly add it to the reading list for my UG extinction lecture/practical.

Lines 140-142: Does a single sentence needs its own sub-heading? Looks a bit weird to me but that might just be my own preference. I’d also add that this is a major point for mass extinctions i.e. they cause a prolonged drop in standing global diversity (i.e. gamma diversity) that is recognisable in the fossil record.

Lines 149-: Some mass extinction workers argue for magnitude calculations rather than rates - as extinction rate calculations might underestimate as presume extinction extended across entire stages e.g. the Rhaetian is ~ 4Myrs long but the ETE lasted around 30Kyrs - a rate calculation assumes the extinction occurred across all 4 million years. See Dunhill et al. 2018AB and Kiessling et al. 2007 in ref list below.

Lines 168-171: I agree with Raup’s statement but I believe it lies in selectivity differences i.e. ecological processes - the difference between the field of bullets or stressor-specific extinction. Dunhill et al. 2018A show distinctly different extinction selectivity between ETE and the background intervals either side despite those 'background' intervals also showing relatively high extinction rates compared to background levels throughout the Phanerozoic. Also, note the geographic range selectivity between background and mass extinction – i.e. Payne and Finnegan 2007.

Lines 206-: RE the Cambrian - I would add that this is where drop in standing diversity becomes important - would it be fair to say that we don't recognise any major mass extinctions in the Cambrian because those high extinction rates are outstripped by even higher origination rates? That said, you get bursts of high extinction but these are accompanied by bursts of even higher origination so you're getting turnover events rather than mass extinction events, which often also go hand in hand with low origination rates (until the recovery kicks in, that is).

Lines 512-: I’d come back to selectivity here again, i.e. that a change in ecological selectivity can be a signal of mass extinction - either a switch to generalistic extinction where selectivity becomes less strong (i.e. with regard to geog range) or stressor-specific extinction signals appear (i.e. reef taxa, benthic taxa, apex predators etc.) – see Payne and Finnegan 2007, Dunhill et al. 2018A, Pimiento et al. 2017,

Cheers,

Alex Dunhill

University of Leeds

References:

Dunhill, A. M., W. J. Foster, S. Azaele, J. Sciberras and R. J. Twitchett (2018A). "Modelling determinants of extinction across two Mesozoic hyperthermal events." Proceedings of the Royal Society B: Biological Sciences 285(1889).

Dunhill, A. M., W. J. Foster, J. Sciberras and R. J. Twitchett (2018B). "Impact of the Late Triassic mass extinction on functional diversity and composition of marine ecosystems." Palaeontology 61(1): 133-148.

Kiessling, W., M. Aberhan, B. Brenneis and P. J. Wagner (2007). "Extinction trajectories of benthic organisms across the Triassic–Jurassic boundary." Palaeogeography, Palaeoclimatology, Palaeoecology 244(1–4): 201-222.

Payne, J. L. and S. Finnegan (2007). "The effect of geographic range on extinction risk during background and mass extinction." Proceedings of the National Academy of Sciences of the United States of America 104(25): 10506-10511.

Pimiento, C., J. N. Griffin, C. F. Clements, D. Silvestro, S. Varela, M. D. Uhen and C. Jaramillo (2017). "The Pliocene marine megafauna extinction and its impact on functional diversity." Nature Ecology & Evolution 1(8): 1100-1106.

---

## [Reviewer Report]

*Comments to Author*: I very much enjoyed reading this version of Prof. Marshall's synthesis as I am sure future readers will. It is an impressive and very educative summary on the topic of mass extinction. The discussion involved in this review was also very interesting and I thank him for the rebuttal comments. I think it is ready and only add a couple more suggestions (completely optional from my perspective), below.

Carl J Reddin

171. Could add that the Norian is nearly 20 myr long?

221-2. Just to check, I think this paragraph might confuse some readers between extinction rate defined per taxon per time unit and extinction rate per taxon without specifying if the rate is assumed to be constant or pulsed over the time bin (my understanding is that calculated rates are generally the latter, with the assumption of tempo being an optional additional step, but we often don't know how fast said pulses might have been). It might be useful to make that distinction for the author's important discussion of comparing temporal rates of extinction (especially as discussed later).

L518. strenghts typo

L544. One could also unnerve the reader by suggesting that both databases may simply suffer from the same biases (e.g. rock exposure patterns)! Also sensu Holland (2020)

652. I still think that other hypotheses as well as the blocking of sunlight are worthy of mention here. The rapid cooling (Brugger et al. 2017) hypothesis with its direct influence on thermal habitat e.g. for dinosaurs (Chiarenza et al. 2020) would also produce a more-rapid-than-modern rate of extinction. So perhaps mention both (or more) rapid mechanisms or leave mechanism open and just mention rapid change associated with bolide impact?

Brugger, J., Feulner, G., Petri, S. (2017): Baby, it's cold outside: Climate model simulations of the effects of the asteroid impact at the end of the Cretaceous. - Geophysical Research Letters, 44, 1, 419-427

Chiarenza, A. A., Farnsworth, A., Mannion, P. D., Lunt, D. J., Valdes, P. J., Morgan, J. V., & Allison, P. A. (2020). Asteroid impact, not volcanism, caused the end-Cretaceous dinosaur extinction. Proceedings of the National Academy of Sciences, 117(29), 17084-17093.

---

## [Reviewer Report]

*Comments to Author*: Thanks for taking on board my comments. I have no further requests to make. I look forward to recommending this as wider reading for my extinction lecture on my level 2 palaeobiology course.

All the best,

Alex Dunhill

---

## [Editor Report]

*Comments to Author*: Dear Prof. Marshall,

Thank you again for submitting your manuscript (ID: EXT-22-0022.R1) to Cambridge Prisms: Extinction. I am pleased to inform you both two reviewers (Carl Reddin and Alex Dunhill) have agreed to accept your manuscript. Dr. Reddin has also made some suggestions that should be useful in improving the clarity of this manuscript.

In addition, I found some minor issues in the references:

1) Line 778, “GSA today” should be “GSA Today”

2) The journal name of PNAS should be uniform in the references, it is “Proc. Natl. Acad. Sci. U. S. A.” in Line 814, but it is “Proc. Natl. Acad. Sci.” in Line 816.

3) Line 843, Capitalized initials of “Extinction, A New Approach” should be eliminated. Similar issues also occur in Lines 858, 873, 997

I think we can finally accept the manuscript after you have addressed these suggestions.

Sincerely,

Haijun

Prof. Haijun Song

Handling Editor, Cambridge Prisms: Extinction

Email: haijunsong@cug.edu.cn

Reviewer 1

I very much enjoyed reading this version of Prof. Marshall's synthesis as I am sure future readers will. It is an impressive and very educative summary on the topic of mass extinction. The discussion involved in this review was also very interesting and I thank him for the rebuttal comments. I think it is ready and only add a couple more suggestions (completely optional from my perspective), below.

Carl J Reddin

171. Could add that the Norian is nearly 20 myr long?

221-2. Just to check, I think this paragraph might confuse some readers between extinction rate defined per taxon per time unit and extinction rate per taxon without specifying if the rate is assumed to be constant or pulsed over the time bin (my understanding is that calculated rates are generally the latter, with the assumption of tempo being an optional additional step, but we often don't know how fast said pulses might have been). It might be useful to make that distinction for the author's important discussion of comparing temporal rates of extinction (especially as discussed later).

L518. strenghts typo

L544. One could also unnerve the reader by suggesting that both databases may simply suffer from the same biases (e.g. rock exposure patterns)! Also sensu Holland (2020)

652. I still think that other hypotheses as well as the blocking of sunlight are worthy of mention here. The rapid cooling (Brugger et al. 2017) hypothesis with its direct influence on thermal habitat e.g. for dinosaurs (Chiarenza et al. 2020) would also produce a more-rapid-than-modern rate of extinction. So perhaps mention both (or more) rapid mechanisms or leave mechanism open and just mention rapid change associated with bolide impact?

Brugger, J., Feulner, G., Petri, S. (2017): Baby, it's cold outside: Climate model simulations of the effects of the asteroid impact at the end of the Cretaceous. - Geophysical Research Letters, 44, 1, 419-427

Chiarenza, A. A., Farnsworth, A., Mannion, P. D., Lunt, D. J., Valdes, P. J., Morgan, J. V., & Allison, P. A. (2020). Asteroid impact, not volcanism, caused the end-Cretaceous dinosaur extinction. Proceedings of the National Academy of Sciences, 117(29), 17084-17093.

Reviewer 2

Thanks for taking on board my comments. I have no further requests to make. I look forward to recommending this as wider reading for my extinction lecture on my level 2 palaeobiology course.

All the best,

Alex Dunhill